# A Machine Learning Framework for Balancing Training Sets of Sensor Sequential Data Streams

**DOI:** 10.3390/s21206892

**Published:** 2021-10-18

**Authors:** Budi Darma Setiawan, Uwe Serdült, Victor Kryssanov

**Affiliations:** 1Graduate School of Information Science and Engineering, Ritsumeikan University, Kusatsu 525-8577, Japan; 2Faculty of Computer Science, Brawijaya University, Malang 65145, Indonesia; 3College of Information Science and Engineering, Ritsumeikan University, Kusatsu 525-8577, Japan; serdult@fc.ritsumei.ac.jp (U.S.); kvvictor@is.ritsumei.ac.jp (V.K.); 4Center for Democracy Studies Aarau (ZDA), University of Zurich, 8001 Zurich, Switzerland

**Keywords:** class-imbalanced data, sensor sequential data, Unrolled GAN

## Abstract

The recent explosive growth in the number of smart technologies relying on data collected from sensors and processed with machine learning classifiers made the training data imbalance problem more visible than ever before. Class-imbalanced sets used to train models of various events of interest are among the main reasons for a smart technology to work incorrectly or even to completely fail. This paper presents an attempt to resolve the imbalance problem in sensor sequential (time-series) data through training data augmentation. An Unrolled Generative Adversarial Networks (Unrolled GAN)-powered framework is developed and successfully used to balance the training data of smartphone accelerometer and gyroscope sensors in different contexts of road surface monitoring. Experiments with other sensor data from an open data collection are also conducted. It is demonstrated that the proposed approach allows for improving the classification performance in the case of heavily imbalanced data (the F1 score increased from 0.69 to 0.72, p<0.01, in the presented case study). However, the effect is negligible in the case of slightly imbalanced or inadequate training sets. The latter determines the limitations of this study that would be resolved in future work aimed at incorporating mechanisms for assessing the training data quality into the proposed framework and improving its computational efficiency.

## 1. Introduction

As modern society increasingly relies on smart technologies, sensors become omnipresent and are used to supply the technologies with ever-growing volumes of data. Many ongoing global initiatives, from Industry 4.0 [1] to IoT and Smart Cities [2], assume collecting data from heterogeneous sensors installed at specific locations that is then subjected to analysis with machine learning algorithms trained to detect and recognize various patterns of interest in the data. At the same time, on the very “local” level of daily life, current and future communication technologies enable an additional layer of data collection—the so-called “human (activity) data” (body movements, physiological parameters, etc.) that are typically obtained from wearable devices, such as smartphones, and used for various purposes [3]. The processing of this data follows the same logic as in the “non-human” case and usually entails finding and classifying data patterns with machine learning algorithms as well [4].

Pattern recognition in machine learning relies on pre-collected “training” datasets which are translated, through learning, to a model that then is used to detect characteristic patterns of the training sets in newly collected data [5]. The learning process implemented with a specific algorithm determines the performance quality of the model; however, that also critically depends on the quality of the pre-collected datasets, i.e., on the representativeness of the training data. A “good” classification model can only be developed with a representative—“large enough and all-inclusive”—training data. Therefore, when building a model for multiple pattern recognition, a major challenge is to assemble a balanced, in terms of samples per pattern of interest (i.e., per class), training dataset. The latter is often difficult to achieve in practice, as many important events, such as, for example, engineering system failure [6], health ailment [7], and hazardous behavior [8], are relatively rare. Sensor signal patterns corresponding to such events would naturally be underrepresented in the data, especially in the case of time-series, resulting in imbalanced training sets. Models built through the learning process with imbalanced data would most likely perform poorly, allowing for reliably detecting patterns corresponding to “normal” or “background” but not rare or less prevalent events. Thus, imbalanced data cause serious problems for the application of popular machine learning algorithms in smart technologies [9].

The study presented in this paper aims to develop a machine learning framework that would allow for addressing the imbalanced data problem. This problem could be resolved, among other possible approaches, either through resampling (i.e., reducing the number of dominant patterns in the training set) or by increasing the number of marginal patterns (i.e., data augmentation by, for example, generating synthetic samples for the underrepresented classes, based on empirical samples available). The presented study focuses on the latter approach to minimize the time and effort required for collecting training sets but also to preserve as much as possible the data representativeness. As most of the sensors used in practice (e.g., for monitoring purposes) produce time-series data [10], the study’s scope is limited to sensor sequential data streams.

The rest of the paper is organized as follows. Section 2 provides a brief overview of related work. Section 3 introduces a framework for data augmentation powered by Unrolled Generative Adversarial Networks (Unrolled GAN) that is the main result of the presented research. The framework is exemplified in a case study described in Section 4 that deals with road surface monitoring, which is an important task in realizing the smart city, smart transportation, and e-government concepts. Then, the proposed approach to data augmentation is additionally tested in several experiments conducted with various data from open data collections, as detailed in Section 5. Section 6 discusses the experimental results obtained and formulates the limitations of the study. Finally, Section 7 concludes the paper.

The main contributions of the presented study are as follows: (1) A novel data augmentation framework for balancing training sets of sensor time-series data has been developed. (2) The framework has been extensively tested in experiments with different datasets, and its limitations have been identified. In addition, the proposed approach has been successfully used to monitor the road surface conditions based on smartphone sensor data.

## 2. Background

### 2.1. Imbalanced Data Classification

A dataset is called imbalanced when the samples it is comprised of are dominated by majority-type event samples, while minority-type events are underrepresented. Machine learning models developed on imbalanced data are typically biased toward majority events and may fail to recognize important events that are rare, sporadic, or irregular, and for that reason, these are underrepresented in the training set [11]. The problem of imbalanced training data is currently well recognized, and it attracted significant attention across multiple research domains [9,12,13].

All existing approaches to addressing training data imbalance that were reported in the literature can be categorized as data level, algorithmic, or hybrid. In an attempt to correct the sample balance, a number of data transformation methods, which could also include some sort of bootstrapping, were developed to modify the training set by adding and removing data (i.e., oversampling or undersampling specific type of events or both) [14,15,16]. It was also found that undersampling can negatively affect the classification performance [17], while oversampling—i.e., data augmentation—is, in many cases, a preferable solution that provides for better results. Compared to the data-level methods, modifying the learning algorithm so as to ensure even participation of all types of events (i.e., classes) in the training process (e.g., see [18]) is a more complex approach that typically requires prior knowledge of data and class distributions. Combining resampling and algorithm adjustments in a hybrid method (e.g., as in [19]) often allows for achieving a good classification performance but also for certain flexibility as the same method would then work reasonably well with different data collections.

### 2.2. Data Augmentation

A variety of data augmentation methods has been proposed to balance training sets of time-series data, such as, for instance, up-sampling or time-stretching, down-sampling, noise addition, rotational distortion, and mirroring [20,21,22,23,24]. To achieve better results, these signal transformation-focused methods are usually used in combination with a machine learning system built around an artificial neural network [25,26]. Among network types and architectures tested, Generative Adversarial Networks (GAN) and their various modifications have often produced the best results [27,28].

Having been originally developed for image processing [29], GAN typically consist of two network models (Figure 1): a generator, G, which transforms a random input vector z into a new synthetic image $x_{g}$, and a discriminator, D, which is used to train the generator. The discriminator receives a real (i.e., non-synthetic) x and synthetic $x_{g}$ image, and it generates an output in the form of a binary decision. The goal set for the discriminator is to prove that $x_{g}$ is a synthetic image, while the generator tries to produce an $x_{g}$ value that would “fool” the discriminator.

A serious limitation (known as “mode collapse” [30]) of GAN is that the generated synthetic data may have low variance and, for that reason, may not be suitable for the inclusion in the training set [31,32]. To reduce the chances of running into the mode collapse and secure the required diversity in the synthetic samples, a modified network structure called Unrolled GAN was proposed [33]. The main idea of Unrolled GAN is to let the generator predict *k*-steps ahead of the discriminator and update the generator parameters based on the predictions. The latter is achieved by training the discriminator *k* steps at each learning iteration to update the generator parameters. The discriminator updates its parameters only once, during the first of the *k* steps.

### 2.3. Augmentation of Sensor Sequential Datastreams with GAN

The application of GAN to augment training sets was, until recently, limited to image data. While there were reports on GAN used to augment non-image data [34] and, more specifically, acoustic data [35,36], such research remains relatively scarce. The authors are also unaware of work that would aim to address the possible mode collapse in a context of time-series data augmentation with GAN. However, given the success of Unrolled GAN in generating high-quality synthetic data, it appears reasonable to expect that this type of adversarial network would be used to improve the classification performance of a machine learning system. The next section presents a general framework developed for the deployment of Unrolled GAN to augment training sets of sensor sequential data streams. Early results of this study have been reported in [37], where synthetic data generated with Unrolled GAN were used without any additional processing steps to augment the training sets described in Section 4.1 of this paper.

## 3. Proposed Approach

Figure 2 gives an overview of the proposed framework for data augmentation that goes through the following steps: training data acquisition, segmentation and sequence labeling, noise addition, sliding window processing, chunk labeling, chunk rearrangement, Unrolled GAN training, synthetic chunk generation, and synthetic data cleaning.

### 3.1. Data Acquisition, Segmentation, and Labeling

Sensor data are sequentially collected and stored in a time-series format. The sensor sampling frequency (or sampling rate) determines the number of numerical values recorded per time unit, and it is set, based on the approximate time of events of interest that are monitored with the data. To detect complex events, one may need to utilize several sensors with their sampling processes being synchronized. As raw data are captured in sequences, data segmentation is required to delineate events of interest when creating training sets. Event labels are (typically manually) assigned to data segments with the corresponding patterns.

### 3.2. Noise Addition

An alternative training set is created from the collected data by adding random noise to the original numerical values in the sequences as follows:(1)$$s^{\prime }=s+\alpha s,$$
where s stands for the data obtained from the sensor,$\;\alpha =\underset{D}{\mathrm{random}}\left[-t,t\right]$, t∈(0,1] defines the noise ratio, D is the distribution of α (uniform by default), and $s^{\prime }$ denotes the noised data. The addition of noise allows for improving the stability of the training process and is intended to contribute to solving the mode collapse problem [38]. This alternative set is also segmented and labeled (and, later, combined with the noise-free labeled data, thus producing a training dataset for the Unrolled GAN).

### 3.3. Sliding Window Processing and Chunks Labeling

The labeled data are arranged into chunks, using a sliding window with a length of l samples and strides of m samples. A chunk includes labeled class sequences, and its own label is determined as:(2)$$chunk\;label=\{\begin{array}{c} \underset{i\in C,\;i\neq 0}{\mathrm{argmax}}\left(q\left(class_{i}\right)\right),\;\;if\;q\left(class_{i}\right)\geq B;\\ \;\;\;0\;\;\;\;\;\;\;\;\;\;\;\;\;\;\;\;\;\;\;\;\;\;\;\;\;\;\;\;\;\;,\;\;\mathrm{otherwise}.\;\;\;\;\;\;\;\;\;\;\; \end{array}$$

Here, $q\left(class_{i}\right)$ gives the number of times sequence label $class_{i}$ appears in the chunk, C={0,1,2,…} defines the set of labels with label $class_{0}$ reserved for the “background” events (such as, for example, silence in voice data, recurrent movements in accelerometer data, etc.) that have the largest representation in the training data. B is a constant, setting the occurrence threshold for a sequence label to become the chunk label.

### 3.4. Chunk Rearrangement

Rearranging chunks is required before feeding them to a classifier system (e.g., see [39]). For the purposes of this study, the labeled sequences are rearranged into two-dimensional data structures which are processed, using two-dimensional convolutional neural networks. However, the definition of a chunk as a sequence is preserved at the conceptual level throughout this paper. For instance, a labeled chunk of 300 sequential values would be converted to a 12×25 two-dimensional data structure that would then be used to train the classifier.

### 3.5. Unrolled GAN Training and Synthetic Chunk Generation

Figure 3 illustrates the concept of n-step Unrolled GAN. The presented GAN structure assumes that in each unrolling step, new discriminator parameters *θ_kD_* are calculated for the ensuing unrolling process. However, only the first calculated parameters *θ_1D_* are used to update the discriminator. To obtain optimal *θ_kD_*, the Adam optimizer [40] is deployed to compute the gradient. The gradient of the last unrolling step is used for updating the generator parameters.

A specific design of the network architecture is obtained by tuning through increasing or reducing the capacity (number of layers and kernels) of the discriminator and generator networks as well as adjusting their learning rates. The tuning objective is to get as close as possible to the Nash equilibrium state, where the generator would produce new synthetic data such that it could successfully “fool” the discriminator. Figure 4 explicates the tuning process that begins with simple networks, which are then gradually expanded. The generator’s capacity is increased or its learning rate is risen if its loss surges considerably at the beginning of learning iterations. Tuning up the discriminator is performed in a complementary manner, as detailed in Figure 4.

In the training process, the generator input z is a vector of random values, whereas the discriminator for its input receives either the output of the generator (i.e., a synthetic sample) or an empirical sample from the training dataset. Before it is fed to the discriminator, each two-dimensional chunk of the empirical sample is scaled to [−1,1]. The training is performed all over the training data for several iterations. After the training process is completed, the generator network is used solely to produce synthetic chunks, from which a synthetic data pool is constructed.

### 3.6. Synthetic Data Cleaning

Chunks in the synthetic pool need to be cleaned before they would be used to balance training sets. This is to eliminate low-quality and “ambiguous” (e.g., when the intended class models overlap) synthetic data. For the cleaning, Phase 1 of the evaluation process proposed in [41] is implemented. A base classifier is trained with a labeled empirical set and run to classify all chunks in the synthetic pool. Then, misclassified chunks are removed from the pool, while the remaining chunks are used for balancing the training sets of sensor data.

## 4. Case Study

There is a growing trend of relying on collections of crowdsourced data in designing and implementing various concepts of smart transportation and smart city [42]. Data recorded with sensors of wearable devices, such as smartphones, can be used to categorize driving behavior [43,44], detect transportation hazards [45], and assess road surface characteristics [46,47]. Smartphone accelerometer, gyroscope, and GPS sensors allow for collecting huge yet typically imbalanced datasets that would be used to train machine learning classifiers for the smart technologies. Below, an attempt is described to deploy the data augmentation framework formulated in Section 3 in the context of the road surface monitoring task.

### 4.1. Task Formulation and Data

Figure 5 depicts the concept of road surface monitoring, using smartphones. Accelerometer and gyroscope sensor data obtained with smartphones carried on vehicles that travel through a road network were analyzed to assess road surface characteristics. It was found that a classifier pretrained on such sensor data allows for localizing potholes, major cracks, joints, and manholes but also speed bumps [48,49]. On the other hand, it was reported that the smartphone sensor-based approach would hardly be used to reliably detect relatively minor surface defects such as longitudinal cracks [50].

To explore the practical applicability of the monitoring method outlined above, a case study was conducted in cooperation with the road maintenance department of the municipal government of Malang, Indonesia (https://dpuprpkp.malangkota.go.id/, accessed on 26 August 2021). Sensor data were gathered with Samsung Galaxy A30 smartphones that were attached to motorcycles in the landscape position, using handlebars. The motorcycles were driven repeatedly at a speed in the range of 25 to 50 km/h through specific roads in the city for a total distance of approximately 10 km, as detailed in Figure 6. The routes were selected based on the presence of road defects and speed bumps. For segmentation, sequence labeling, and validation purposes, images of the road surface were recorded with cameras also attached to the motorcycles. Camera and smartphone timestamps were synchronized every time before making a trip.

To collect the smartphone sensor data, an Android application was developed and installed on the smartphones (see Figure 7). The application recorded the accelerometer and gyroscope data at 50 Hz in each of the three dimensions (sensor data indexed with x, y, and z), and stored it in the CSV format.

The collected data sequences were segmented and labeled manually by the authors, based on recorded images of the corresponding routes. After consultations with road maintenance specialists and taking into account results of the previous studies [48,49,50], it was decided to limit the inspection to four road surface conditions with class labels assigned as follows: “flat road” (Class 0), “pothole” (Class 1), “speed bump” (Class 2), and “bumpy road” (Class 3).

A noised (alternative) dataset was created, using the raw data, as specified by Equation (1) with t=0.1. The data of this set were segmented and labeled in the same manner as the noise-free data.

### 4.2. Sliding Window, Chunk Labeling, and Rearrangement

A sliding window with l=50 and m=10 was used. The selected window length corresponds to the number of samples recorded per one second, as most road surface events in focus would be sensed within a second when driving at a speed in the specified time interval. The sliding distance was selected so as to accommodate for cases when a segment with an event of interest spans over two chunks. All chunks were labeled, as prescribed by Equation (2) with threshold B set to 20.

The chunks were rearranged into two-dimensional vectors that could be fed to the two-dimensional input layer of the network system (described in detail in the next subsection). The x, y, and z values of the accelerometer and gyroscope data were first stored in a one-dimensional sequence that was then folded into a two-dimensional vector of the 12×25 size, as illustrated in Figure 8.

With a total of 3179 chunks thus obtained, an empirical data pool was created that contained 2695 chunks labeled Class 0, 189 chunks labeled Class 1, 96 chunks labeled Class 2, and 199 chunks labeled Class 3. The pool was split into training (80%) and testing (20%) sets for the classifier models. As expected, Class 0 events were (naturally) dominant, and the training data required balancing for Class 1, Class 2, and Class 3 events.

### 4.3. Unrolled GAN Network Design and Synthetic Chunk Generation

Figure 9 and Figure 10 present final designs of the generator and discriminator networks, respectively, which resulted from Unrolled GAN tuning performed by the authors. The learning rates were set at $5\times 10^{-4}$ for the generator and $1\times 10^{-4}$ for the discriminator. These values were determined in the course of the tuning process as well.

The input of the generator G (see also Figure 3) is a one-dimensional vector of 100 values randomly drawn from the standard normal distribution. The first hidden layer of the network is fully connected, while the second is a 512×3×7 three-dimensional deconvolution layer. All the subsequent hidden layers are deconvolution layers, and the output of this network is a two-dimensional 12×25 vector representing a new synthetic data chunk. The output layer uses the hyperbolic tangent, tanh(·) as the activation function (it is expected to have output values in the range of [−1,1]), while all other layers, except for the input and output, use the Leaky Rectified Linear Unit function (Leaky ReLU [51]) for activation. All deconvolution layers have a kernel size of 3×3 and the first two layers use a stride of 2×2 before connecting to the next layer.

The discriminator D receives data from the empirical pool but also from the output of G arranged in a two-dimensional 12×25 vector. All the hidden layers of the discriminator network are convolution layers, and the last layer is flattened into a fully connected layer. The output of D is a binary unit with State 0 to indicate synthetic data, and State 1 indicates empirical data inputs. Binary cross-entropy is utilized as the loss function. Each layer, except for the input and output, uses Leaky ReLU for activation, while the output layer uses the sigmoid function. All convolution layers have the 3×3 kernel size, and the first two layers use a 2×2 stride before connecting to the next layer.

For each of the Class 1, Class 2, and Class 3 events, the Unrolled GAN were trained separately, performing 10,000 training iterations with results recorded every 1000 iterations. In all cases, the training process was completed in 10 unrolled steps, as suggested in [33]. Then, a synthetic data pool was generated with approximately 3000 synthetic chunks per class after cleaning. For data augmentation, synthetic chunks with the corresponding class label were selected randomly and added to the training data to balance the underrepresented events (i.e., the events of Class 1, Class 2 and Class 3).

### 4.4. Base Classifier

A base classifier system was developed for analyzing road surface conditions based on the sensor data collected with smartphones. This system was also used to examine the quality of data generated by the Unrolled GAN, to clean the synthetic data, and to assess the effect of data augmentation on the classifier system performance.

Figure 11 depicts the structure of a deep convolutional neural network (DCNN) utilized as the base model in the study. The network has four hidden convolution layers and one fully connected layer. For the activation function, all hidden layers use a rectified linear unit (ReLU) [52], and the output layer uses the softmax function. Adam is used for the optimizer, the categorical cross-entropy is used for the loss function, and the initial learning rate is set at 0.01. The input is a two-dimensional 12×25 vector, while the output is the per class-label probability.

### 4.5. Evaluation of Data Augmentation with the Unrolled GAN

Figure 12 displays results of evaluating (with the base classifier model) synthetic data produced by the Unrolled GAN in 1000 training iteration increments. As one can see from the figure, precision and recall tend to slightly fluctuate over the iterations, and it is not obvious when the training process should be stopped. On the other hand, recall for Class 2 stays rather low throughout. However, the latter would be expected, given the relatively small size (77 chunks) of the corresponding training set (manual inspection revealed that synthetic chunks with this label were frequently misclassified as Class 3 events).

It turned out that monitoring the loss functions of the generator and discriminator networks provides for better criteria in managing the training process (Figure 13). In particular, by observing changes in the average and the variance of the generator loss, one could identify the range of training iterations when the Nash equilibrium was maintained (also see [53,54]). For instance, the average loss does not increase or increases slowly, and the loss variance is relatively low within each of three intervals bounded with the red dashed lines in the graphs. Then, it could be expected that the Unrolled GAN produced synthetic data of an acceptable quality when operated in this range. Therefore, in the presented study, synthetic chunks generated after 6000, 4000 and 4000 iterations were used for training data augmentation of Class 1, Class 2, and Class 3 events, respectively.

Figure 14 presents results of road surface assessment with the base classifier trained on the data collected with the smartphone sensors. The results obtained for Class 0 (background/flat road) events suggest that the developed classifier would be an adequate system for the given task. The effect of training data augmentation on the classifier performance is especially noticeable for Class 1 and Class 2 events. There is also an improvement for Class 3 events; however, this would still be difficult to detect reliably with the given system.

### 4.6. Exploring Possibilities for Data Crowdsourcing

In reality, there is a huge number of different contexts determined by, among other factors, sensor and smartphone models used, transportation means, and smartphone orientation and placement, in which accelerometer and gyroscope sensor data would be recorded. Taking into account this fact, as well as the relatively poor classification performance obtained for the Class 3 events, it was decided to extend the case study with an additional data collection scenario and also to modify the classification task as follows.

A new dataset was collected using Asus Zenfone Max Pro smartphones. Motorcycles were driven repeatedly through the routes shown in Figure 6. Before each trip, a smartphone with the data collection application (Figure 7) installed was put in the driver’s shirt pocket in the portrait position, with the phone’s screen facing the driver’s body. Cameras, whose timestamps were synchronized with timestamps of the smartphone sensors, were also installed on the motorcycles to record visual information. The raw data collected were preprocessed in the same way as described in Section 4.2, creating an empirical data pool of 5970 chunks. Then, the data were manually classified into “flat road” (Class A, 3835 chunks), “road anomaly” (Class B, 2041 chunks), and “human body movement” (Class C, 94 chunks) events. The training and data augmentation processes were conducted in the same manner as described in Section 4.4 and Section 4.5, with the output layer of the base classifier being suitably adjusted to accommodate for the new classification structure. The results of this experiment are presented in Figure 15.

As one can see from the figure, data augmentation with the Unrolled GAN allowed for achieving a nearly 70% recognition rate for the “road anomaly” events that are typically events of interest for road maintenance departments of municipal governments. While the change of the data collection context (i.e., smartphone model and placement) had, apparently, no major influence on the anomaly detection, the generalization of the road condition definitions made the effect of the data augmentation practically more useful than in the case of the Figure 14 experiment.

## 5. Experiments with Other Sensor Data from Open Collections

The general framework for training data augmentation described in Section 3 was also tested with various sensor time-series data from the UCR Time Series Classification Archive [55]. Table 1 outlines two experiments conducted with relatively small datasets from the archive, which were recorded with sensors of different types. Table 2 and Table 3 summarize the experimental results obtained (in the tables, the bold and italic fonts are used to highlight changes, increases and decreases, respectively, in the classification performance after data augmentation). As one can see, the selected training sets are heavily imbalanced and very small in size compared to the testing sets, which mimics a sensor data monitoring context. However, in both cases, training data augmentation with the Unrolled GAN led to an increased classification accuracy, as indicated with the corresponding values of F1 score observed. (Note that the data of Table 1 were used as benchmark sets, i.e., fixed and complete sets with no random context.).

To explore how augmenting a relatively large and slightly (as opposed to heavily) imbalanced data would affect the classification performance, an experiment with the “FordA” set from the UCR Time Series Classification Archive was conducted (see Table 4). The data were randomly split into five folds, and the effect of data augmentation was assessed for each fold. The F1 scores averaged over the classes and folds stay at 0.945 and 0.950 for the imbalanced and balanced data, respectively. The observed difference is not statistically significant (p=0.16). Therefore, it can be presumed that the application of the proposed approach would be hard to justify in the case of large but slightly imbalanced training sets. 

## 6. Discussion

### 6.1. Comparison with Existing Approaches

Results of the experiments described in Section 4 and Section 5 have convincingly demonstrated that the proposed data augmentation framework allows for improving the classification performance of the base classifier system working with sensor sequential data-streams. Changes in the performance are especially noticeable in the first experiment of the case study, where the heavily imbalanced yet naturally noisy training data were used (Figure 14). The perceived difficulty of this classification task makes it interesting to look into how other related approaches would perform with the data. Figure 16 compares the effects of balancing the training set (Section 4.2) with three popular time-series data augmentation methods (noise addition, time stretching, and undersampling; see [19,20,21,22,23]) and the Unrolled GAN (as proposed in this study) on the classification performance. Since time stretching and undersampling could not produce enough synthetic data when used alone, each of these methods was complemented with noise addition in the experiment. In all evaluation scenarios, the testing data were used identically.

It is evident from Figure 16b that balancing the training data merely with noise resulted in performance drops for Class 1, Class 2 and Class 3 events. Time stretching plus noise addition led to an increased performance for Class 1 events but also to performance drops for Class 2 and Class 3 (Figure 16c). Many chunks with Class 2 events were misclassified as Class 3 and those with Class 3 events were misclassified as Class 1 in this case. Undersampling with noise addition (Figure 16d) could apparently not produce synthetic data of an acceptable quality, as it only led to a decreased recall for Class 2. On the other hand, the Unrolled GAN-powered data augmentation resulted in increasing the classification performance for all but Class 0 (background) events (Figure 16e).

### 6.2. The Roles of Synthetic Data Cleaning and Noise Addition

Among problems associated with data augmentation powered by GAN, the inability of the networks to consistently generate synthetic data of an acceptable quality is frequently cited [56]. Along with the mode collapse, there are two other major sources of this problem: non-optimal early stopping of the learning process [57] that would result in the “premature” generation of low-fidelity synthetic data and, more fundamentally, a low-diversity, unrepresentative training sample translated to an overfitting generator model [38]. To assess the stability of the proposed approach when the distribution of events of interest over the training and testing sets changes, chunks of the empirical data described in Section 4.2 were reshuffled, producing four additional pairs of training and testing sets. Synthetic data were generated with the Unrolled GAN trained on the additional sets, and these were used to augment the corresponding training data fed to the base classifier. The experiment was repeated with and without data cleaning and noise addition. The trained classifier performance was evaluated with the relevant test sets. Figure 17 presents results obtained with two additional sets (selected for the illustrative purposes and denoted Fold 2 and Fold 3) in comparison to the performance achieved with the data of the case study (Fold 1). In all three cases, training data were augmented with the Unrolled GAN but was not subjected to cleaning and noise addition. As one can see from the figure, the augmented training sets are consistently associated with better recall for all underrepresented (i.e., Class 1, Class 2, and Class 3) events but Class 3 of Fold 3. However, at the same time, there are significant drops in precision that are especially evident in the case of the underrepresented events of Fold 2.

By manually inspecting the synthetic data of Fold 2, it was found that there were multiple instances of prematurely generated chunks. Figure 18 shows examples of successfully (a) and prematurely (b) formed synthetic chunks of Class 2.

Figure 19 illustrates the effect of synthetic data cleaning (Section 3.6; the base classifier was trained on the same data that were used to train the Unrolled GAN) and noised data addition (Section 3.2, t=0.1) in the case of Fold 2. As one can infer from the figure, these two steps allow for obtaining better results for a majority of the classes, if not all. Furthermore, the observed tendency is quite stable: when evaluated with the data of all five folds, gains in recall were registered in 13 of the 20 trials (5 folds × 4 classes), and gains in precision were registered also in 13 trials. The F1 score averaged over the classes and folds stands at 0.69 for the imbalanced training sets and at 0.72 for the augmented (Unrolled GAN + cleaning + noised data) sets (p=0.007). Therefore, it can be said that the proposed data augmentation framework is robust in respect to variations in the training–testing data configuration.

### 6.3. Limitations and Future Work

While this is a fact that balancing training sets through data augmentation can boost the classifier system performance, results can vary significantly, depending on the augmentation method used but also on the training data available. No data augmentation method could help when the training data are incomplete or are too noisy with respect to the events of interest. For example, the six samples of Class 0 events of the SonyAIBORobotSurface1 dataset were apparently not representative enough to build a reasonably accurate classifier model even with the balanced data (see Table 2). The same can be said about the Class 3 training set of the first experiment of the case study (Figure 14). Unrolled GAN-powered data augmentation is quite computation-intensive, but it does not offer mechanisms and metrics for assessing the training data quality. This, together with the necessity of GAN manual tuning, is seen as the main limitation of the described approach.

The proposed framework would require modifications to deal with heavily skewed or fat-tailed distributions of events in training sets. More specifically, distributions D other than the uniform one would be used to noise the data, along with controlled (non-uniform) sampling to add synthetic data to underrepresented classes in the training set. However, the data used in this study’s experiments did not require such modifications.

Adjusting the GAN training procedure would help reduce the chances of premature chunk generation and, therefore, increase the quality of synthetic data. Analyzing the training dynamics would, on the other hand, help estimate the quality of training data. This, together with work on improving the computational efficiency of the data augmentation framework, determines the future directions of the presented study.

## 7. Concluding Remarks

The main contribution of the study presented in this paper is the developed data augmentation framework for balancing training sets of sensor sequential data streams (time-series data) intended for machine learning classifiers. The centerpiece of the framework is the Unrolled GAN system. The framework was extensively tested with data collected from smartphone accelerometer and gyroscope sensors in different contexts of the road surface monitoring task but also with three different datasets from the UCR Time Series Classification Archive. Through the experiments conducted in the study, it was demonstrated that the proposed approach allows for achieving a better classification performance of a base classifier trained on augmented sets, is robust to changes in training/testing data configurations, and outperforms such popular augmentation methods as noise addition, time stretching, and undersampling. It was also shown that GAN would hardly be a “silver bullet” (if there would ever be one) that would single-handedly solve the training data imbalance problem. Rather, Generative Adversarial Networks should be deployed as part of a multi-step approach, such as, for example, the one presented in this paper, to successfully deal with class-imbalanced datasets of sensor sequential data.

## Figures and Tables

**Figure 1 sensors-21-06892-f001:**
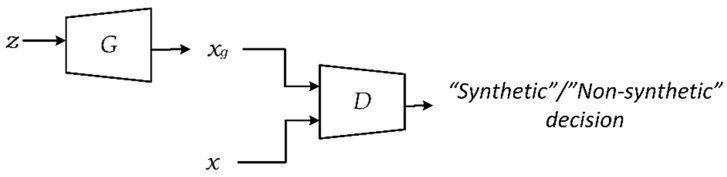
GAN basic architecture.

**Figure 2 sensors-21-06892-f002:**
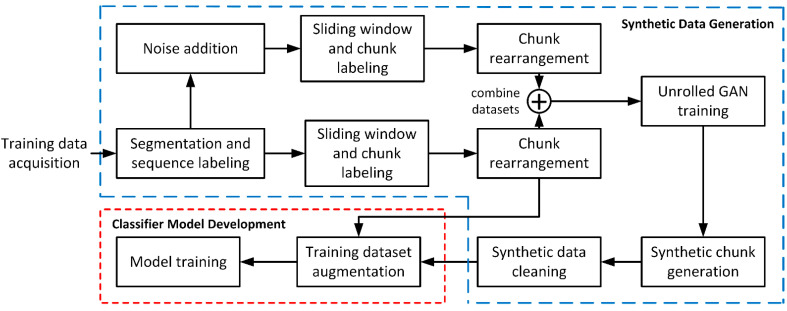
An overview of the proposed approach.

**Figure 3 sensors-21-06892-f003:**
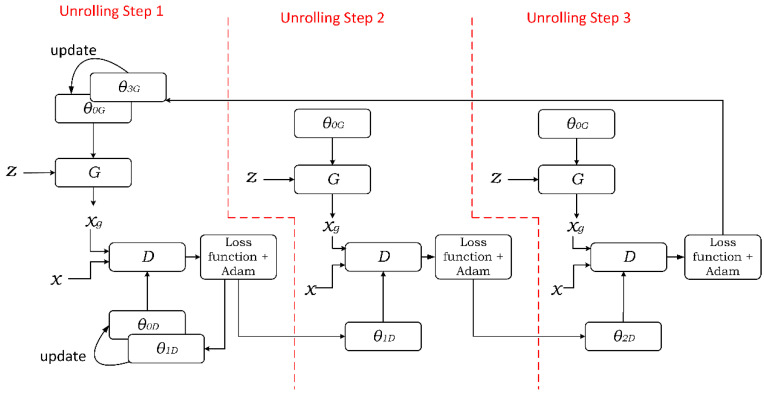
n-step Unrolled GAN for data augmentation. For the sake of clarity, consideration in this figure is limited to the case of n=3, with z denoting a random vector and x standing for the empirical data vector (input), $x_{g}$ representing the synthetic data vector produced by generator G, D standing for the discriminator, $\theta_{0G}$ representing the initial parameter vector of G, $\theta_{3G}$ representing the generator parameters for the three unrolling steps, $\theta_{0D}$ representing the initial parameter vector of D, and $\theta_{kD},\;k\in \left\{0,1,n-1\right\}$ signifying discriminator parameters at the kth unrolling step.

**Figure 4 sensors-21-06892-f004:**
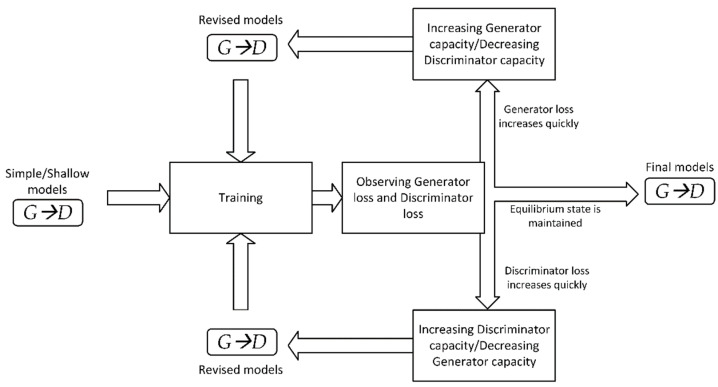
Unrolled GAN tuning.

**Figure 5 sensors-21-06892-f005:**
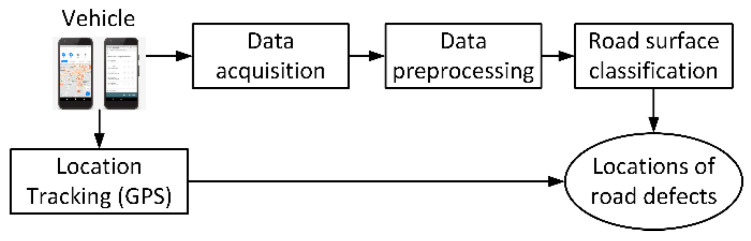
Road surface monitoring with smartphones.

**Figure 6 sensors-21-06892-f006:**
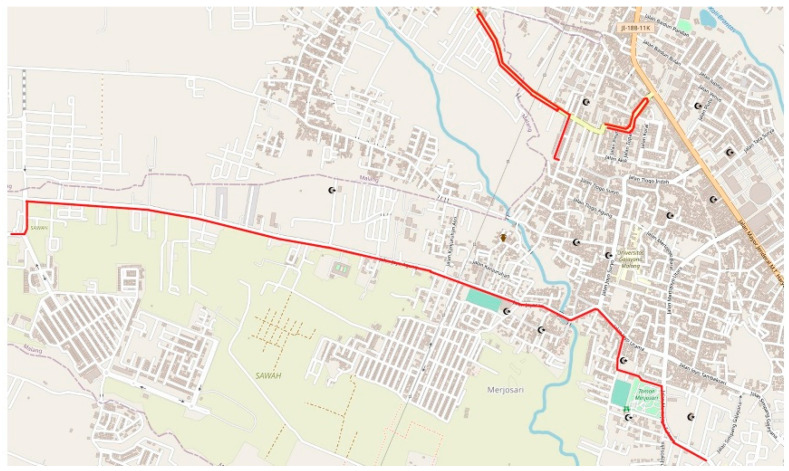
Data acquisition routes (marked with red) in Malang, Indonesia. (Map data copyright by OpenStreetMap contributors; available from https://www.openstreetmap.org/#map=16/-7.9383/112.5941, accessed on 14 March 2021).

**Figure 7 sensors-21-06892-f007:**
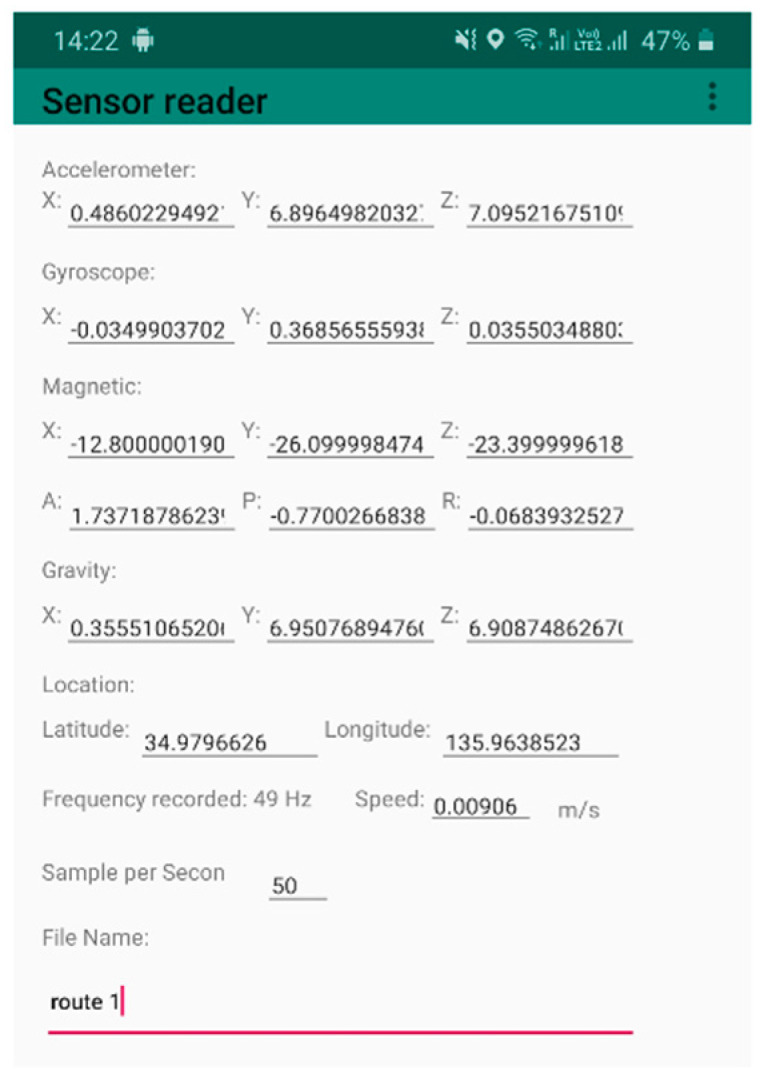
A screenshot of the android application developed for data acquisition.

**Figure 8 sensors-21-06892-f008:**
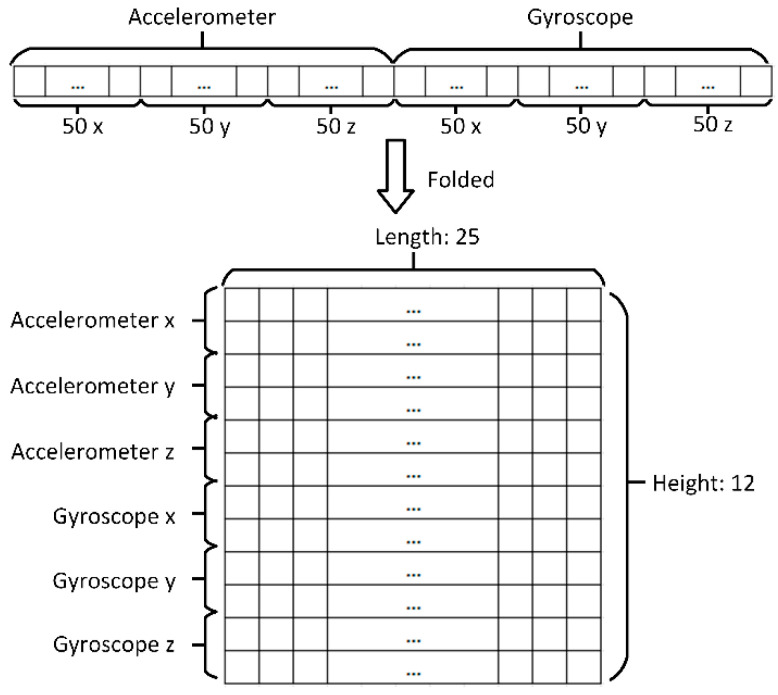
Data rearrangement: a one-dimensional sequence (1×300) is folded into a two-dimensional vector (12×25).

**Figure 9 sensors-21-06892-f009:**
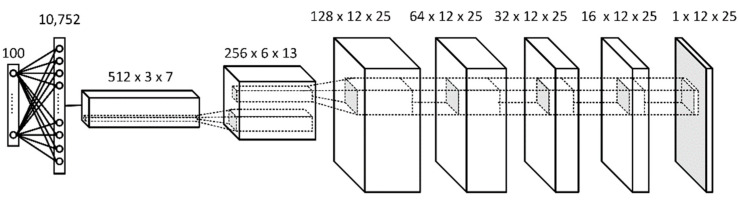
Generator G of the Unrolled GAN used in the study.

**Figure 10 sensors-21-06892-f010:**
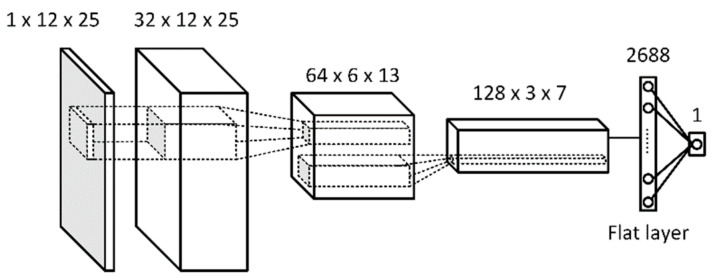
Discriminator D of the Unrolled GAN used in the study.

**Figure 11 sensors-21-06892-f011:**
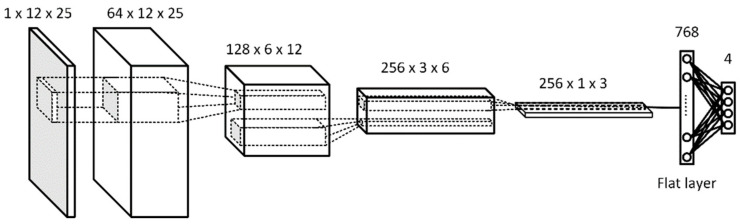
The base DCNN model for the classifier.

**Figure 12 sensors-21-06892-f012:**
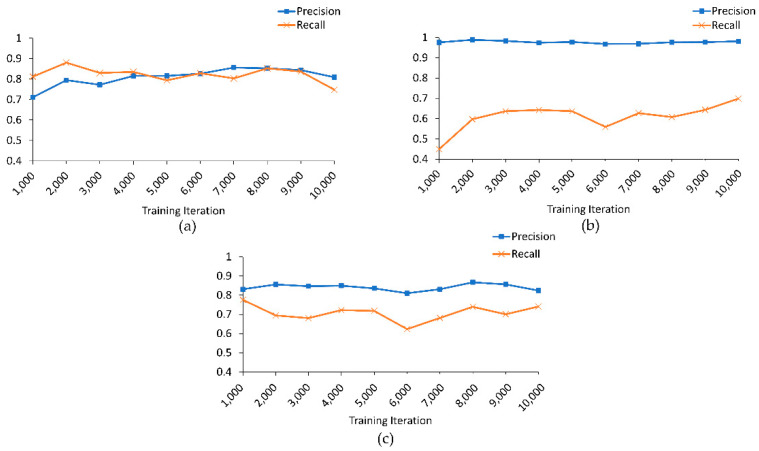
Precision and recall calculated for the synthetic data of (**a**) Class 1, (**b**) Class 2, and (**c**) Class 3.

**Figure 13 sensors-21-06892-f013:**
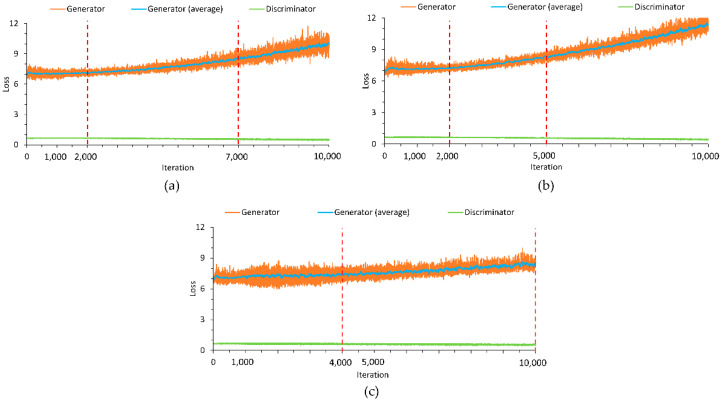
Generator and discriminator loss values recorded during the training process for (**a**) Class 1, (**b**) Class 2, and (**c**) Class 3 events.

**Figure 14 sensors-21-06892-f014:**
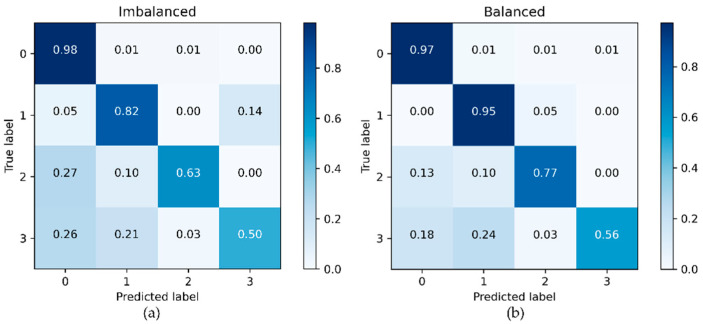
Confusion matrices for road surface assessment with the base classifier trained on (**a**) the original (imbalanced) set and (**b**) the original data augmented with the Unrolled GAN.

**Figure 15 sensors-21-06892-f015:**
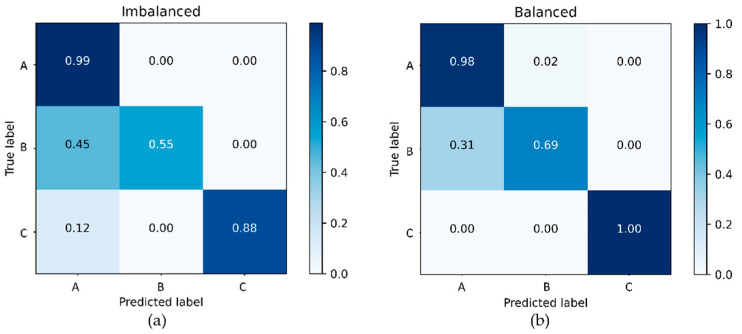
Confusion matrices for road anomaly monitoring with the base classifier trained on (**a**) the original (imbalanced) set and (**b**) the original data augmented with the Unrolled GAN.

**Figure 16 sensors-21-06892-f016:**
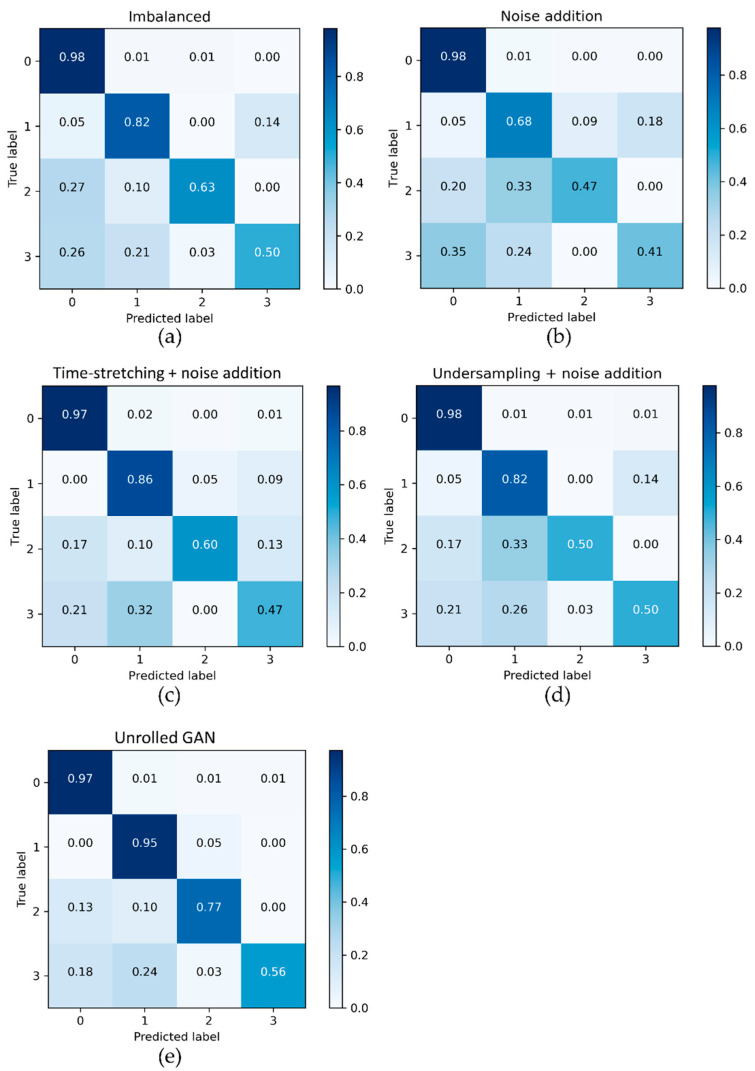
Classification performance of the base classifier trained on: (**a**) unbalanced data, (**b**) noise addition augmented data, (**c**) time-stretching + noise addition augmented data, (**d**) undersampling + noise addition, and (**e**) the augmented data of the case study.

**Figure 17 sensors-21-06892-f017:**
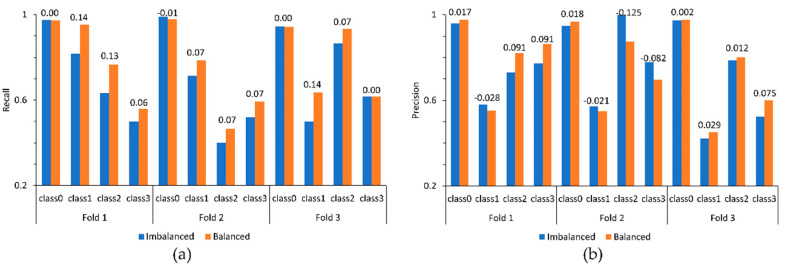
Changes in recall (**a**) and precision (**b**) observed for three different compositions of training and testing sets (figures on the top of the bars are the corresponding differences; see the main text for details).

**Figure 18 sensors-21-06892-f018:**
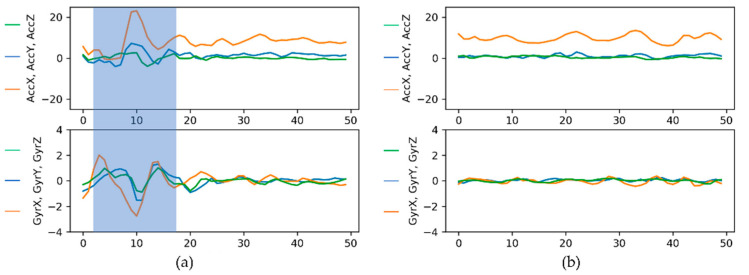
Class 2 synthetic chunks generated by the Unrolled GAN: (**a**) successfully formed (the signal pattern corresponding to a speed bump is highlighted) and (**b**) prematurely generated.

**Figure 19 sensors-21-06892-f019:**
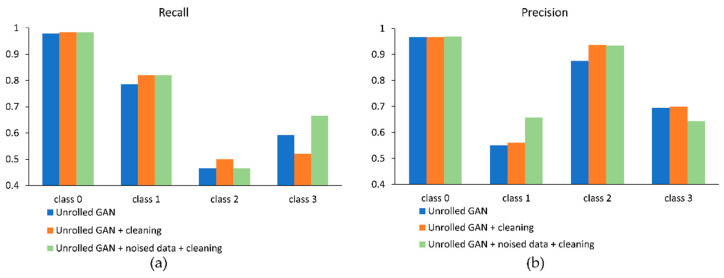
Recall (**a**) and precision (**b**) of the base classifier before and after data cleaning and noised data addition for Fold 2.

**Table 1 sensors-21-06892-t001:** Data from the UCR time-series classification archive used in the experiments.

1	Dataset Name	SonyAIBORobotSurface1
	Training data size	20
	Testing data size	601
	Time-series length	70
	Training data events	Class 0: 6; Class 1: 14
	Description	*x*-axis accelerometer data of an Aibo robot walked on two different surfaces (concrete and carpet)
2	Dataset Name	CinCECGTorso
	Training data size	40
	Testing data size	1380
	Number of classes	4
	Time-series length	1639
	Training data events	Class 0: 5; Class 1: 13; Class 2: 12; Class 3: 10
	Description	ECG data

**Table 2 sensors-21-06892-t002:** The effect of training set augmentation for the SonyAIBORobotSurface1 data.

	Imbalanced (Original) Training Set	Training Set Balanced with Unrolled GAN
Precision	[0.90, 0.57]	[*0.89*, **0.59**]
Recall	[0.48, 0.93]	[**0.52**, *0.91*]
F1 score	[0.62, 0.71]	[**0.66**, **0.72**]
F1 score (average)	0.66	**0.69**

**Table 3 sensors-21-06892-t003:** The effect of training set augmentation for the CinCECGTorso data (the precision, recall, and F1-score values are ordered by class).

	Imbalanced (Original) Training Set	Training Set Balanced with Unrolled GAN
Precision	[1.00, 0.98, 0.99, 0.78]	[1.00, **1.00**, *0.98*, **0.82**]
Recall	[0.72, 0.99, 0.99, 1.00]	[**0.78**, *0.98*, **1.00**, 1.00]
F1 score	[0.83, 0.99, 0.99, 0.88]	[**0.87**, 0.99, 0.99, **0.90**]
F1 score (average)	0.92	**0.94**

**Table 4 sensors-21-06892-t004:** Configuration of the *k*-fold cross-validation experiment with the FordA data from the UCR Time Series Classification Archive.

Dataset name	FordA	
Number of classes	2	
Time-series length	500	
Fold data size	Training:	Class 0: 2022; Class 1: 1915
	Testing:	Class 0: 505; Class 1: 479
Number of folds, *k*	5	
Description	Engine noise (acoustic data)	

## Data Availability

Data is available at https://github.com/s-budidarma/dataset-smarphone-sensor-for-road-surface-monitoring/tree/main/motorcycle, accessed on 16 September 2021.

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
