# Peer review of "A Machine Learning Framework for Balancing Training Sets of Sensor Sequential Data Streams"

_sensors, 2021, doi:10.3390/s21206892_

Round 1

Reviewer 1 Report

The paper is well-written, clear, and well-structured. The problem setting is well explained and there are plenty of experimental results. The novelty is limited because the general framework presented in section 3, is an adaptation to an already published one. The section 3 figures are really very expressive! The results are clearly described  and interpreted with the necessary “nuances” and critical attitude. 

**** The results described should be seen as coming from work in progress. Therefore I would suggest to add to section 6.3 on “limitations” an extra paragraph on HOW the authors would tackle the mentioned limitations and bottlenecks in future work.

Some more details:

Abstract: add a sentence on the obtained results (quantitative) and the main bottlenecks that  require more work

Section 1:

Lines 57-63: Split the phrase “While … representativeness” -- >

While this problem would -- > This problem could

samples available) the presented -- > samples available). The presented

Line 87: would -- > could ##?##

Line 89: type events -- > type of events

Line 94: drop “,to be implemented,"

Line 97 strange use of “when” ??

**** Line 135: mentions [35], it would be good to describe in one or two sentences how the current results in the new paper augment those of [35], because in my opinion the results described in the current paper are still not conclusive and produced by work that is still in progress.

Line 202: in the figure -- > in figure 4

Table 1  Accustic -- > Acoustic

Table 1 Why only two classes instead of 11 are used for the Acoustic data set?

**** Tables 2, 3 and 4: statistical testing of the significance of the differences is required. There might be differences in numbers  but as long as the differences are not significant, for me the performance is equal.

Figure 17: for the sake of clarity, add the meaning of the colors

Reviewer 2 Report

  1. The dataset size of Table 1 is not large enough. Please use much larger datasets to verify the proposed idea.
  2. It is better to summarize the contribution in the section of abstraction, such as "This work makes the following contributions: (1) ... , (2) .... and (3) ...
  3. Some recent related publications may be added as references, such as

[1] J. Zheng, et. al., "Improving the Generalization Ability of Deep Neural Networks for Cross-Domain Visual Recognition", IEEE Transactions on Cognitive and Developmental Systems, vol. 13, no. 3, pp. 607-620, 2021.

[2] Shorten, C., Khoshgoftaar, T.M. & Furht, B. Text Data Augmentation for Deep Learning. J Big Data 8, 101 (2021). https://doi.org/10.1186/s40537-021-00492-0.

Reviewer 3 Report

In their work, the authors propose an Unrolled Generative Adversarial Networks (Unrolled GAN)-powered framework to tackle the problem of training data imbalance resulting from sensor sequential data streams. Overall, the contribution is well written, easy to follow and presents a valid and quite comprehensive evaluation of the proposed method/framework. I also like the Limitations section (6.3) towards the end of the manuscript, which discusses current shortcomings that still need to be addressed by the community (or the authors) in future research works. In addition, I like that the authors provide their data on GitHub to the community (even if the link is not directly clickable in the manuscript, I had to assemble it together myself. Hence, please check if the link is working directly in the final version of your manuscript):
https://github.com/s-budidarma/dataset-smarphone-sensor-for-road-surface-monitoring/tree/main/motorcycle

In summary, I endorse the publication of the manuscript.
